# Astrocytic TRPV4 Channels and Their Role in Brain Ischemia

**DOI:** 10.3390/ijms24087101

**Published:** 2023-04-12

**Authors:** Jana Tureckova, Zuzana Hermanova, Valeria Marchetti, Miroslava Anderova

**Affiliations:** 1Institute of Experimental Medicine, Czech Academy of Sciences, 1083 Videnska, 142 20 Prague, Czech Republicmiroslava.anderova@iem.cas.cz (M.A.); 2Second Faculty of Medicine, Charles University, 84 V Uvalu, 150 06 Prague, Czech Republic

**Keywords:** TRPV4, Ca^2+^ signaling, ischemia, glia, astrocytes

## Abstract

Transient receptor potential cation channels subfamily V member 4 (TRPV4) are non-selective cation channels expressed in different cell types of the central nervous system. These channels can be activated by diverse physical and chemical stimuli, including heat and mechanical stress. In astrocytes, they are involved in the modulation of neuronal excitability, control of blood flow, and brain edema formation. All these processes are significantly impaired in cerebral ischemia due to insufficient blood supply to the tissue, resulting in energy depletion, ionic disbalance, and excitotoxicity. The polymodal cation channel TRPV4, which mediates Ca^2+^ influx into the cell because of activation by various stimuli, is one of the potential therapeutic targets in the treatment of cerebral ischemia. However, its expression and function vary significantly between brain cell types, and therefore, the effect of its modulation in healthy tissue and pathology needs to be carefully studied and evaluated. In this review, we provide a summary of available information on TRPV4 channels and their expression in healthy and injured neural cells, with a particular focus on their role in ischemic brain injury.

## 1. Introduction

During cerebral ischemia, rapid increases in intracellular calcium ([Ca^2+^]_i_) initiate dramatic changes in the nervous tissue, leading to cell death and reactive gliosis. Transient receptor potential (TRP) channels of the vanilloid subfamily, including TRPV1, TRPV2, and TRPV4, are of great interest as they play important roles in ischemia-induced excitotoxicity, inflammation, and apoptosis [1]. In this review, we focus on the TRPV4 channel, which was first described as an osmosensitive channel in the kidney [2] yet is also widely expressed in the nervous system, namely in neurons [3,4,5], astrocytes [6,7,8], endothelial cells [9], smooth muscle cells [10], microglia [11,12] and in cells of the oligodendroglial lineage [13]. In the pathology of cerebral ischemia, TRPV4 is involved in both harmful and protective processes. On one hand, neuronal TRPV4 channels contribute to excitotoxicity via enhancement of the N-methyl-D-aspartate (NMDA) receptor function [14,15,16], yet on the other hand, endothelial and astrocytic TRPV4 are involved in the positive regulation of cerebral blood flow and thus the improvement of hypoperfusion in the affected area [10,17,18,19,20]. In addition, astrocytic TRPV4 channels are key players in the development of cerebral edema, and their role in this process is protective [21,22]. Finally, TRPV4 channels also support the protective function of microglia and their ability to migrate to the site of damage [12]. Therefore, TRPV4 is often highlighted as a potential target for the treatment of cerebral ischemia. To date, researchers have introduced several prospective TRPV4 effectors (agonists and antagonists), some of which can be administered orally [23], and the phenotype of TRPV4 knockout mice further suggests that its therapeutical inhibition may be without side effects. However, results from experiments modulating the TRPV4 function in animal models of cerebral ischemia are highly inconsistent. It is therefore questionable, whether this multifunctional channel, which most likely plays a role in both protective and detrimental processes, is indeed a promising therapeutic target. This review aims to summarize the available data on the role of TRPV4 in different cell types of the central nervous system (CNS) during ischemic brain injury, with the focus on astrocytes.

## 2. TRPV4 Channels

TRPV channels are a subfamily of a larger class of proteins, the TRP channels, which are divided, according to DNA and the protein sequence homology, into six groups: the TRPV (V for vanilloid), TRPA (A for ankyrin), TRPC (C for canonical), TRPM (M for melastatin), TRPML (ML for mucolipin) and TRPP (P for polycystin). These channels are widely expressed throughout the whole body and in the CNS. We can highlight three families, namely TRPC, TRPM, TRPV (reviewed by Samanta et al., 2018 or Zhang and Liao 2015 [24,25]). A TRPV subfamily comprises six members, TRPV1-TRPV6. In this review, we focus on one member of this family, the TRPV4, which is a particularly interesting Ca^2+^ permeable channel in terms of its contribution to the pathology of cerebral ischemia. In humans, TRPV4 is expressed on the cell membranes in various tissues. According to the Human Protein Atlas, these tissues include the brain (mainly the cortex and hippocampus), endothelium, skin, urinary and respiratory systems, heart, spleen, liver, duodenum, pancreas, adrenal gland, testes, and placenta (https://www.proteinatlas.org/ENSG00000111199-TRPV4/tissue; accessed on 5 September 2022). In the CNS, its expression has been described in neurons, glia, and endothelial cells [9,26].

The structure of the TRPV4 channel is tetrameric, and every subunit consists of six transmembrane domains (TM1-6), with the pore formed by the interaction between TM5 and 6. The protein consists of 871 amino acids with amino (N) and carboxyl (C) termini, both located intracellularly (Figure 1). Most of the protein structure (70%) is exposed intra- or extracellularly and contains numerous binding sites for interaction with modulators and proteins [10,27]. The N-terminus represents most of the cytosolic portion of TRPV4, and it includes the ankyrin repeat domain (ARD) that distinguishes TRPV and TRPC channels from TRPM. ARD is an important site for protein-protein interactions [28,29], and is probably responsible for the N-termini self-association into the tetrameric structure [30]. The so-called proline-rich domain (PRD) is located near the ARD, which is important for the mechanosensitivity of the TRPV4 channels [31]. In addition, phosphorylation sites for the protein kinases C (PKC), cAMP-dependent–phosphorylation site, or Src family-dependent tyrosine phosphorylation site, have been localized in the N-terminus [32]. The C-terminus of TRPV4 contains binding sites for interaction with some proteins, such as the microtubule-associated protein 7 (MAP7) [33], inositol triphosphate (IP3) receptor [34], and calmodulin (CAM) [35].

The important role of ARD and its interactions is additionally accentuated by the fact that numerous naturally occurring point mutations in this part of the protein lead to the development of serious genetic disorders, such as spinal muscular atrophy (SMA), hereditary motor and sensory neuropathy type 2 (HMSN2) or spondylometaphyseal dysplasia (SD). In general, these diseases are characterized by bone malformations and their abnormal growth, causing disproportions in the skeleton, such as scoliosis or a short trunk [36,37]. Additionally, other symptoms such as muscle weakness, movement problems, scoliosis, or breathing difficulties occur in SMA patients. Lack of balance and reflexes and deficits in hearing, sight, and feeling are connected to HSMN2, and a short trunk or abnormalities in the pelvis are characteristic for SD. Disease-causing mutations quite often occur in ARD. However, the same effect occurs also, thanks to mutations, in the pore region or other domains such as the C-terminus. All these point mutations affect the basal activity of TRPV4, but none of them causes a complete loss of function [37,38].

TRPV4 is a polymodal non-selective cation channel, mainly permeable for Ca^2+^, Mg^2+^ or Na^+^, characterized by a relatively high Ca^2+^ permeability ratio (PCa^2+^/PNa^+^ = 6–10, PMg^2+^/PNa^+^ = 2–3) [28]. The ion selectivity is determined by two aspartate residues (Asp672 and Asp682) located in the pore-forming loop of the channel between TM5 and 6. The substitution of either or both of these aspartates with alanine led to the reduction of the relative permeability for divalent cations, as well as reduced outward rectification [39]. The Ca^2+^ current may be inward as well as outward rectifying, depending on the concentrations in the intra- and extracellular space (ECS) [40]. A single channel conductance ranges between the values 90–100 pS for outward currents and 50–60 pS for inward currents [41,42,43]. These values can change because of the disease-causing point mutations mentioned above. Some of the naturally occurring mutations lead to a lesser TRPV4 conductivity, whereas with others the conductivity can be increased [37].

TRPV4 channels can be activated by a broad range of stimuli, including physical (cell swelling [44], heat [43], mechanical stimulation [45], endogenous ligands (endocannabinoids, arachidonic acid (AA), and 4-α-phorbol esters) and exogenous/synthetic ligands (reviewed by Everaerts et al., 2010, or Nilius et al., 2003 [27,28]).

### 2.1. Osmo-/Mechanosensation

The osmosensitivity of TRPV4 channels was the first characteristic that was described. They were originally called osmosensitive transient receptor potential channel-4 (OTR-PC4) or vanilloid receptor-related, osmotically activated channel (VR-OAC), until later studies showed that they could also be activated by other physical and chemical stimuli. TRPV4 is well known to be activated by hypoosmotically induced cellular swelling [2], and the activation may occur either directly or via an indirect pathway of cellular modulators (reviewed by Toft-Bertelsen and MacAulay [46]). Direct mechanical activation of the channel may be attained by the connection of its intracellular part to cytoskeletal components such as actin, microtubules, and microfilaments with a specific binding site for F-actin in the TRPV4 N-terminus [47,48]. The indirect pathway includes the membrane-stretch-induced activation of phospholipase A2 (PLA2), leading to the production of AA which is in turn metabolized by cytochrome P450 (CYP450) epoxyoxidases to epoxyeicosatrienoic acids (EETs) [20].

### 2.2. Activation by Heat

TRPV4 has also shown activation by heat, like other members of the TRPV family. However, compared to TRPV1, 2 or 3, which are activated by temperatures over 30 °C, TRPV4 is activated by moderate heat, i.e., at temperatures >24–27 °C [41,49]. The mechanism of activation is not yet fully understood, but it is likely to be indirect activation by other thermosensitive molecules, similar to swelling activation. This has been proven, for example, by the fact that heat activation disappears during inside-out patch clamp recordings [28,43]. However, contrary to swelling activation, these molecules are not CYP450 or PLA (and PLA-dependent AA formation), which has been proven by using blockers of these two molecules [44].

### 2.3. Endogenous Ligands

Endogenous TRPV4 ligands mainly include endocannabinoids, such as anandamide, AA, and its derivatives [20]. This pathway for TRPV4 activation involves the metabolism of AA to EETs mediated by CYP450 epoxygenase [44,50]. EETs, on the one hand, cause vasodilation of the arteries due to the hyperpolarization of smooth muscle cells [44,51,52], and on the other hand, they can directly interact with TRPV4 channels and activate them. The putative binding site for EETs is located at the N-terminus of the TRPV4 channel [53].

In contrast, phosphatidylinositol 4,5-bisphosphate (PIP2) can be considered an endogenous antagonist of TRPV4 channels. Although some members of the TRP family require PIP2 for their activation, it is inhibitory for TRPV4 [54,55]. The interaction between PIP2 and TRPV4 occurs at the ARD [29].

### 2.4. Exogenous Agonists

There are several different exogenous TRPV4 channel agonists among which GSK1016790A (PubChem CID 23630211) and 4α-Phorbol-12,13-didecanoate (4αPDD) are probably the most used (reviewed by Vincent and Duncton [56]). Notably, the former is up to 300-fold more potent than the latter [57]. A small molecule, phorbol ester 4αPDD is a selective TRPV4 agonist, which, at low concentrations (ED50 > 25 μM) does not need PLA and AA, and its effect is therefore direct [41,53,58]; however, it requires a Tyrosine-555 and Serine-556 in the TM3 of the TRPV4 channels [44]. GSK1016790A is a potent and selective agonist, patented by the GlaxoSmithKline (GSK) company. It has been broadly utilized in in vivo pharmacological studies of TRPV4 activation [57,59,60,61]. Another known agonist that has already been used in several in vitro and in vivo studies [56,62] is RN-1747 (PubChem CID 5068295), a benzenesulfonamide, uncovered by knowledge-based screening of a library of commercial agents. In addition to these widespread compounds, a novel agonist, a derivative of quinazolin-4(3H)-one, has recently been identified [63].

The extract of Andrographis paniculata, containing Bisandrographolide A, which has been shown to selectively activate TRPV4 [64], is also of particular interest.

### 2.5. Exogenous Antagonists

There are several non-selective TRPV4 antagonists, including trivalent cations Gd^3+^, La^3+^ [65], Capsazepine [66] or ruthenium red [41]. However, the demand for inhibitors with greater specificity has led to the testing of new compounds. Novel TRPV4 antagonists HC-067047 from Hydra Biosciences (PubChem CID 2742550) and RN-1734 (PubChem CID 3601086) from Renovis, were isolated from the same focused library of commercial aryl sulfonamides as TRPV4 agonists RN-1747 or RN-9893. Additionally, GlaxoSmithKline company published several molecules antagonizing TRPV4 channels, all showing structural similarities to the GSK1016790A agonist. The most important of these is GSK2193874 (PubChem CID 53464483).

Citral, a major component of lemongrass, frequently used as a taste enhancer, is noteworthy among the natural substances acting as TRPV4 channel antagonists [67].

## 3. Non-Glial TRPV4 Channels in the Brain

In the CNS several cell types express the TRPV4 channels. The heterogenous population of glial cells will be discussed later, but aside from glia, TRPV4 can also be found in neurons or endothelial cells. Thanks to their wide distribution throughout the brain parenchyma and a broad range of activating stimuli, TRPV4 channels are suspected to contribute to numerous diseases, such as ischemia [68,69], glioblastoma [70], intracerebral hemorrhage [71], Parkinsonism [72] or even depression [73]. In the following sections, we discuss the role of TRPV4 channels in specific cell types with the focus on their role in the cerebral ischemia.

### 3.1. Neurons

Regarding neurons, several authors described the correlation between the TRPV4 activation and functions and other neuronal channels, such as TRPV1, Neuronal calcium sensor 1, Calcium-activated potassium channels, or neurotransmitter receptors, and these interactions then affect the functions of TRPV4 channels, such as osmosensing [3,15,16,74,75]. The TRPV4 channels are activated by physiological temperature, and this activation leads to a slight depolarization of the resting membrane potential, as was shown in hippocampal neurons [76,77,78].

Moreover, TRPV4 channels and their activation affect neurotransmitter receptors and their expression in hippocampal neurons [16,79]. The relation between TRPV4 channels and glutamate receptors plays a role in the ischemic brain injury, when a TRPV4-mediated increase in the function of NMDA receptors contributes to excitotoxicity and subsequent tissue damage [14,15,16]. Additionally, the activation of TRPV4 is further enhanced by hyperthermia and can contribute to the development of a brain edema after cerebral ischemia, which again further enhances tissue damage [80]. On the contrary, the activation of TRPV4 channels decreases the size of the brain infarction after an ischemic stroke, as reported by Chen and colleagues, and, in addition, it leads to stronger neurogenesis [81]. Prolonged exposure to heat stress also causes a decrease in the expression of TRPV4 channels [74]. Therefore, it seems that the activation of TRPV4 channels during cerebral ischemia is detrimental for the mature neurons in the affected area, but it subsequently helps to replace the damaged neurons with new ones.

### 3.2. Endothelial Cells

In the brain, TRPV4 channels are also expressed on the membranes of endothelial cells (ECs). Interestingly, they show interspecies differences. Luo and colleagues reported numerous roles and functions of TRPV4 channels in rat ECs, compared to human cells [82]. However, even in human ECs, the activation of TRPV4 channels was reported to affect the proliferation of ECs, their migration and formation of capillaries within the brain, as well as in the retina [83,84]. In the adult brain, the activity of TRPV4 channels is, under physiological conditions, silenced by the PIP2, and the inhibition increases endothelial resistance by affecting endothelial intercellular contacts [85,86]. If activated (for example, via AA and CYP450), TRPV4 contributes to vasodilation and a decrease in the blood pressure by affecting endothelial intercellular contacts [18,19,20]. However, under pathological conditions, the activation of TRPV4 channels can cause a switch of the ECs to an inflammatory phenotype and is therefore damaging to the ECs in the brain and spinal cord [86,87,88]. Additionally, pathological conditions can cause over-expression of TRPV4 channels in endothelial cells and their subsequential activation disrupts the integrity of blood-brain-barrier (BBB) via affecting the cytoskeleton and promoting stress fiber formation [71].

## 4. TRPV4 Channels in Glial Cells

The TRPV4 channels are expressed by several groups of glial cells, including astrocytes and microglia. These cells possess a wide range of functions vital for the healthy functioning of the brain, but they are also crucial in CNS pathologies. In this study we summarize available information on TRPV4 channel expression in glial cells and how this expression affects glial functions in health and disease.

### 4.1. Microglia

Microglia are resident macrophages of the CNS and, therefore, possess various functions associated with health and disease. As the CNS immune cells, microglia communicate with other cell types, scan their environment, and respond to inflammatory signaling and other indications of a disease [89]. Several members of the TRP protein family, such as TRPM, TRPC, and TRPV channels, were described on the membranes of microglia. From the TRPV subfamily, the TRPV4, together with TRPV2 and TRPV1 channels, were detected; however, they were all found in cultured microglia [11,12]. On the contrary, the expression of TRPV4 channels on microglial membranes in vivo is believed to be very low at best [90], and this discrepancy in the expression of TRPV4 channels complicates any attempt to elucidate their possible roles in microglia.

The protective role of microglia in the CNS is dependent on their ability to move. The movement of microglia, especially of their processes, is temperature-dependent and affected by the loss of TRPV4 channels. If the TRPV4 channels are missing from microglia, the temperature-dependent movement of their processes is lost in physiological as well as in pathological conditions [12]. The movement of microglial processes is affected by TRPV4 channels because of their permeability to Ca^2+^. When activated, TRPV4 channels are highly permeable for Ca^2+^, which then interacts with microtubules and actin and affects cytoskeletal dynamics, promotes actin polymerization, and, therefore, affects cellular movement [47,48]. Additionally, microglia with acute inhibition of TRPV4 channels show changes in morphology. However, when TRPV4 is knocked out, the changes do not occur [91].

During pathological conditions, the body often responds with fever, during which the activation of TRPV4 can be enhanced; consequently, the movement of microglia is also increased to help the cells scan their environment and deal with the pathology [12]. Moreover, TRPV4 channels also function as mechanical sensors in pathological conditions, and their activation is enhanced during cellular swelling. The subsequent increase of intracellular Ca^2+^ functions as a detector of homeostatic imbalance. The increase in [Ca^2+^]_i_ is usually considered pro-inflammatory (not only in microglia, but in macrophages in general) and is associated with changes in microglial morphology and metabolism, and with the switch towards the activated state [90,92,93].

Contrary to this belief, Konno and colleagues reported that the silencing of TRPV4 activity leads to the attenuation of microglial activation in response to lipopolysaccharide. They used TRPV4 antagonists or TRPV4 knockout to block the effect caused by the TRPV4 activation [94]. To summarize, there is a discrepancy in the literature about the expression of TRPV4 channels in microglia, which raises questions about their functions in the intact brain. The available evidence points to opposing effects of the TRPV4 channel activation on microglial cells, pro- as well as anti-inflammatory, and the evidence so far is therefore inconclusive and needs to be further addressed.

### 4.2. NG2 Glia and Oligodendrocytes

NG2 cells are a very heterogenous subpopulation of glial cells with a vast proliferative potential that gives rise to oligodendrocytes (based on which they are also termed oligodendrocyte precursor cells (OPCs)), astrocytes, or astrocyte-like cells [95,96,97,98]. In general, there is limited knowledge of the TRPV4 channels in NG2 glia, but their expression has been demonstrated in cultured cells as well as in vivo in neonatal and adult animals [13]. However, Kirdajova and colleagues demonstrated that the level of expression of TRPV4 channels varies between the different subpopulations of NG2 glia, which occur after CNS injuries. The highest numbers of cells expressing TRPV4 were reported in “oligodendrocyte-like” NG2 cells and “astrocyte-like” NG2 cells, whereas the subpopulations with the highest expression of NG2 markers showed much lower numbers of TRPV4-positive cells [96]. TRPV4 channels are Ca^2+^ permeable; therefore. we can speculate that their activation increases [Ca^2+^]_i_ and participates in intracellular Ca^2+^ signaling, which in NG2 glia can affect a great number of cellular functions, such as the motility or axon myelination [99,100]. This has been proven by Liu and colleagues, who reported that the deletion of TRPV4 channels prevented cuprizone-induced demyelination by affecting mitochondrial functions [101]. Overall, there is some evidence about TRPV4 channels being expressed in NG2 cells and oligodendrocytes in healthy tissue as well as after a CNS injury; however, the functions of these channels in the physiological and pathological conditions remain to be elucidated.

### 4.3. Astrocytes

Astrocytes comprise a large and heterogeneous group of glial cells that perform various essential functions in the CNS. They play a role in brain development and metabolism, control the CNS microenvironment, modulate synaptic transmission and neurotransmitter release [102], and have an important function in controlling systemic circulation and maintaining cerebral blood flow [103,104]. Astrocytes are also crucial for providing the structural and functional integrity of the nervous system. The astrocyte processes are strategically positioned to make numerous contacts with neuronal synapses, on one side, and brain capillaries on the other. In this manner, they mediate communication between neuronal cells and the vascular system and sense changes in neuronal synaptic activity.

Astrocytic TRPV4 have been described to be involved in neurovascular coupling, neuronal excitability, cell volume control, and regulation of brain blood flow [105]. The activation of astrocytic TRPV4 channels induces an outward rectifying cation current and intracellular Ca^2+^ signals dependent on the presence of extracellular Ca^2+^ [7,106]. Elevated [Ca^2+^]_i_ levels can activate Ca^2+^-activated ion channels, Ca^2+^-dependent intracellular enzymes, or additional Ca^2+^ release from internal stores. These mechanisms will be explained in more detail in the following paragraphs.

#### 4.3.1. Astrocytic TRPV4 Expression

The expression of the TRPV4 channel in the brain astrocytes has been analyzed in several studies, but the results vary considerably, both in the amount of expressing cells and in the level of cellular expression. TRPV4 appears to be restricted to a specific population of astrocytes, and the number of expressing cells differs between brain regions. This channel has been detected in astrocytic cultures isolated from rodent brains. In primary rat astrocyte culture, Benfenati and co-authors verified TRPV4 expression immunocytochemically and recorded an outwardly rectifying cation current activated by 4αPDD, with biophysical and pharmacological properties that overlapped with those of recombinant human TRPV4 expressed in COS cells [7]. They also reported neuronal and astrocytic expression in the rat cerebral cortex based on the immunohistochemical staining, and the enrichment of the channel in astrocytic processes of the superficial layers of the neocortex and endfeet facing pia and blood vessels. Consistent with this study, Kim and co-authors [8] found TRPV4 expression on astrocytic perivascular endfeet in rat cortical brain slices, colocalized with glial fibrillary acidic protein (GFAP), a widely used astrocytic marker, and with aquaporin (AQP4), a typical astrocytic water channel. Hippocampal TRPV4 immunoreactivity in GFAP positive cells has been described by Bai and co-authors [6] in organotypic slices and slices from juvenile rats. They also detected TRPV4 positivity using the Western blot method. On the contrary, Toft-Bertelsen and co-authors [107] did not find any TRPV4 positivity in the rat hippocampal homogenate (P21) or in the astrocyte-enriched fraction (P20–P23) using the Western blot analysis. A difference between these two studies may lie in the loading quantity of the lysate, which was 4-fold lower in the study of Toft-Bertelsen. Methodological differences aside, it is likely that astrocytes in general contain small amounts of TRPV4 protein, and that only particular subpopulations of astrocytes express it. Quantification of TRPV4 positivity performed by immunohistochemistry and in situ hybridization in the hippocampus of adult C57Bl6 mice revealed only 30% of GFAP-positive and 20% of S100β-positive cells expressing TRPV4 [106]. The authors described a subset of astrocytes that express the TRPV4 channel and respond to neuronal signals. Importantly, these cells activate neighboring TRVP4-negative astrocytes via ATP and gap junction-mediated signaling. Therefore, the activation of TRPV4 channels also changes [Ca^2+^]_i_ in TRVP4-negative cells. Even less astrocytic expression of TRPV4 channels was observed using methods of gene expression profiling. Only 3% of the cells expressing TRPV4 were detected in the cortex of GFAP/EGFP mice (P30, P90; sorted based on GFAP- or glutamate aspartate transporter (GLAST)-positivity) by single-cell RT-qPCR [68]. Similar results were obtained by RNAseq in cortical astrocytes (P7; sorted based on aldehyde dehydrogenase 1 family member L1 (Aldh1l1)-positivity) [108], in adult striatal and hippocampal astrocytes (P63; sorted based on Aldh1l1-positivity) [109], as well as in uninjured and reactive spinal cord astrocytes (P56-70; sorted based on ACSA2-positivity) [110]. One issue that should be considered when interpreting single-cell PCR or RNAseq data may be the treatment of cell suspensions prior to mRNA isolation, which may result in the loss of locally expressed proteins as well as mRNA, predominantly in cellular processes and astrocytic endfeet [111,112]. Interestingly, a hundred percent of Müller glia, a specific astrocytic population of retina, express TRPV4 channel [113].

Similar to the above-mentioned studies, Diaz and co-authors observed TRPV4 expression in only a minority of GLAST-positive cells in the cerebral cortex, but this number was significantly increased in hypertension [114]. This finding is in agreement with the idea that TRPV4 expression can change in response to various pathological stimuli, such as ischemia [115]. In the rat hippocampus, Butenko and co-authors showed that astrocytic TRPV4 expression increased after global ischemic brain injury induced by two-vessel carotid occlusion. This observation was documented immunohistochemically and functionally by the patch-clamp technique and Ca^2+^ imaging [115]. Increased TRPV4 expression 1 and 7 days after ischemia coincides with the astrogliosis development. The role of TRPV4 in astrocyte activation during ischemia was also proposed by Yi and co-authors [116].

#### 4.3.2. Channels Cooperating with TRPV4 in Astrocytes

Several channels in the astrocytic plasma membrane located in the vicinity of TRPV4 channels are possibly linked in a functional complex sensing changes in neuronal activity, cellular metabolism, and ionic homeostasis. These include mainly AQP4, glutamate receptors and some types of potassium channels, specifically inwardly rectifying (Kir), large conductance Ca^2+^-activated (BK_Ca_), and ATP-sensitive (K-ATP).

The main astrocytic water channel AQP4 is expressed in membranes facing pia and brain capillaries. It is responsible for bidirectional water transport through the astrocytic membrane on the vascular site as well as through the astrocytic syncytium [117]. Its colocalization and the correlation of its expression with TRPV4 channels have been described in Müller glia and astrocytes [113,118]. Moreover, TRPV4 is likely responsible for the increase in AQP4 expression in the astrocyte membrane following hypothermia [119] or exposure to hypoxic insult [120]. This process is activated by intracellular Ca^2+^ binding to CAM, which subsequently activates AQP4 phosphorylation, causing its relocalization to the astrocyte membranes (Figure 2A). Benfenati and co-authors described the formation of the AQP4/TRPV4 complex in astrocytes, which is essential for regulatory volume decrease (RVD) under hypoosmotic conditions. On the contrary, Jo and co-authors did not observe the formation of a similar complex in retinal Müller glia, but they proposed a mechanism for the interaction of these two channels and their concerted cooperation [113]. According to their suggestion, hypoosmotic stress causes water influx through AQP4 channels, hence cellular swelling. Membrane stretch subsequently activates the TRPV4 channel, which leads to a Ca^2+^ entry and triggers RVD (Figure 2B). The interaction of TRPV4 with other types of AQP has also been described [121,122,123].

Inwardly rectifying K^+^ channel subunit 4.1 (Kir4.1) is the main astrocytic inwardly rectifying channel, which is involved in the potassium uptake and spatial buffering and contributes to the astrocyte swelling under conditions of high extracellular K^+^ concentrations [124]. Similarly to AQP4 and TRPV4, it is enriched in astrocytic endfeet, and its interaction with AQP4 and the formation of complexes have been described [125]. The cooperation of AQP4 and Kir4.1 channel is mainly involved in the process of spatial buffering, whereby excess K^+^ ions, followed by water molecules, are transferred to regions with lower concentrations [126]. Additionally, a correlation between the expression of AQP4, Kir4.1, and TRPV4, suggesting that these proteins work in synergy, has been observed in retinal Müller cells [113]. In the proposed model, Kir4.1 channel activity is suppressed by AA, which is a product of the activity of the stretch-sensitive PLA2. However, such a correlation was not proven in brain astrocytes [22,68].

Other K^+^ channels that are linked with TRPV4 in location and activity are BK_Ca_. They are voltage- as well as intracellular Ca^2+^-dependent and their activation leads to a massive efflux of K^+^ and subsequent membrane hyperpolarization [127]. Their cooperation with TRPV4 has been described mainly in endothelial and smooth muscle cells, where they are crucial for the regulation of blood flow through cerebral vessels in response to changes in neuronal activity [10]. A similar mechanism probably operates in astrocytes, which also contribute to vasodilation of cerebral blood vessels [128]. However, activation of TRPV4 and subsequent increased perfusion can paradoxically cause vasogenic edema and worsening of damage, as shown in traumatic brain injury [129]. Moreover, activation of BK_Ca_ channels by elevated cellular Ca^2+^ levels is also implicated in the RVD process, as suggested by Jo and co-authors for Müller glia [113].

Other potassium channels which are, beside Kir4.1, involved in the astrocytic potassium buffering are K-ATP channels. These channels are activated by either a decrease in cellular ATP levels or an increase in ADP levels, and thus open in response to ischemia/hypoxia or metabolic failure. Their activation leads to hyperpolarization of cells, thus linking cellular metabolism to membrane excitability [130]. Hyperpolarization of the neuronal membrane inhibits Ca^2+^ influx into cells via voltage-dependent channels and consequently reduces excessive glutamate release and subsequent excitotoxicity [131,132]. The activation of K-ATP therefore has the opposite effect on neurons to TRPV4 activation, which enhances excitotoxicity. Since mechanosensory Ca^2+^ responses in astrocytes mediated by interaction between TRPV4 and Cx43 channels lead to release of ATP [133], it is reasonable to speculate that TRPV4 can block the activity of K-ATP channels. In astrocytes, the opening of K-ATP channels modulates communication by gap junctions [134,135] and positively regulates glutamate uptake via glutamate transporters [136,137]. K-ATP are composed of Kir6.1 and Kir6.2 pore-forming subunits and the sulfonylurea receptors SUR1, SUR2A and SUR2B, specifically Kir6.1 and SUR1 in astrocytes [135]. Like TRPV4 and other potassium channels, its activity may be modulated by AA and its metabolites [138,139]. These channels in astrocytes have a SUR1 subunit, the same as SUR1-TRPM channels, key players in brain edema formation [140], and thus are targets for common channel modulators such as glibenclamide, which has an anti-inflammatory effect in the CNS disorders and prevents brain edema [139,141].

In addition to potassium channels, it is necessary to mention the cooperation of TRPV4 channels with glutamate transporters, which occurs mainly in neurons, where TRPV4 activation promotes the activity of ionotropic glutamate receptors [14,15,16]. However, astrocytic TRPV4 channels also work in concert with glutamate receptors, specifically metabotropic glutamate receptors (mGluRs). This cooperation functions in neurovascular coupling (NVC), where astrocytic mGluRs respond to increased neuronal activity and are responsible for the [Ca^2+^]_i_ increase triggering the synthesis of AA metabolites [17]. This process is described in more detail in the following section.

**Figure 2 ijms-24-07101-f002:**
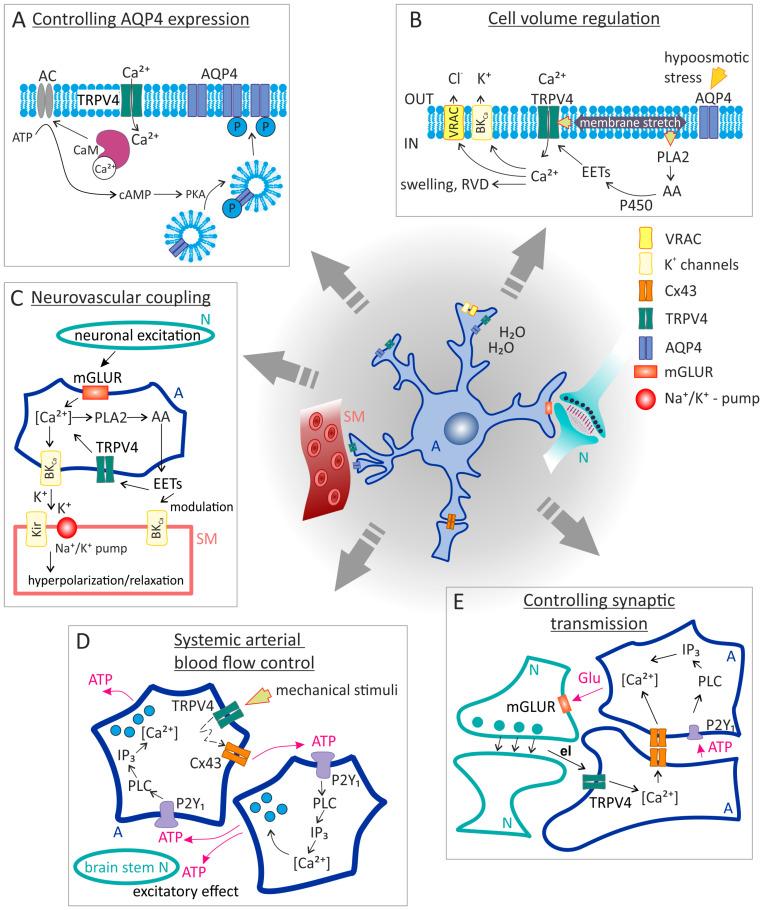
Basic functions of astrocytic TRPV4 channels. (**A**) Influence of AQP4 expression in astrocyte membrane: Ca^2+^ entering the cell through TRPV4 channels binds to CaM, which subsequently activates AQP4 phosphorylation, causing its re-localization to the astrocytic membranes. (**B**) Cell-volume regulation: The mechanism involves cooperation between the AQP4 and TRPV4 channels. The water influx via AQP4 causes astrocyte swelling, which activates the mechanosensitive TRPV4 channel. Subsequent entry of Ca^2+^ triggers RVD via BK_Ca_ or VRACs. Additionally, the membrane stretch can activate PLA2, whose activity leads to the production of AA and its subsequent conversion to EETs. EETs can then further stimulate TRPV4 channels. (**C**) Neurovascular unit: Astrocytic mGluRs sense the extracellular glutamate released from neurons due to their increased activity. The activation of mGluRs leads to the Ca^2+^ influx and following stimulation of BK_Ca_ releasing K^+^ into the ECS. Extracellular K^+^ activates Kir channels in SM cells, causing vasodilation. In addition, PLA2 activation leads to the production of EETs, the opening of TRPV4 channels, and an additional Ca^2+^ increase in astrocytes. (**D**) Systemic blood flow control: Mechanical stimulation of TRPV4 opens the channel for Ca^2+^ entry into astrocytes and activates the interaction between TRPV4 and Cx43 hemichannels. Subsequent release of ATP into the ECS can trigger activation of astrocytic P2Y1 or neuronal purinergic receptors of brain stem sympathetic control circuits. Moreover, astrocytic P2Y1 activation enhances astrocytic excitation by stimulating the further Ca^2+^ release from internal stores. (**E**) Control of synaptic transmission: Activation of astrocytic TRPV4 channels by the endogenous ligand leads to excitation of neighboring astrocytes, even those that do not express TRPV4, either via gap junction mediated Ca^2+^ waves or by ATP release. These astrocytes then release glutamate, which binds to mGluRs on synaptic neurons and thus modulates the neurotransmission. Abbreviations: A, astrocyte; AA, arachidonic acid; AC, adenylyl cyclase; AQP4, aquaporin 4; ATP, adenosine triphosphate; BK_Ca_, Ca^2+^-dependent K^+^ channels; CaM, calmodulin; ECS, extracellular space; EETs, eicosanoid metabolites; el, endogenous ligand; Kir, inward-rectifier potassium channel; mGluRs, metabotropic glutamate receptors; *N*, neuron; P2Y1, G-protein coupled purinergic receptor P2Y1; PKA, protein kinase A; PLA2, phospholipase 2; RVD, regulatory volume decrease; SM, smooth-muscle cell; VRAC, volume-regulated anion channel. The figure was inspired by schematics in publications by Kitchen et al., 2020 [120], Jo et al., 2015 [113], Turovsky et al., 2020 [133] and Shibasaki et al., 2014 [106].

#### 4.3.3. Controlling Cerebral Blood Flow

Astrocytes contribute to the regulation of cerebral blood flow either at the level of the NVC, in response to increased neuronal activity [17,142,143], or from directly sensing the brain perfusion pressure [103,133]. Neurovascular coupling is defined as alterations in local cerebral blood flow in response to changes in neuronal activity. It is mainly mediated by either K^+^ [142,144] or nitric oxide (NO) signaling pathways [145]. Both pathways respond to increased neuronal activity and lead to vasodilation of cerebral arterioles. Signal transmission within NVC involves astrocytic Ca^2+^ signaling, which can be triggered by the activation of mGLURs [146] or the TRPV4 channels. The mGLUR activation in astrocytes induces rises in [Ca^2+^]_i_ which in turn trigger the synthesis of AA and its metabolites such as EETs or prostaglandin-2 (PGE2), both possessing the vasodilatory effect [104,146]. These can both either act directly on the smooth muscle cells, or they can modulate astrocytic K^+^ channels and influence K^+^ release into the ECS [147,148]. In addition, EETs activate TRPV4 channels, and this activation leads to a further increase of Ca^2+^ in astrocytes. A direct link between TRPV4 channel activation, local Ca^2+^ oscillations in astrocytic endfeet, and the regulation of cerebral microcirculation in response to neuronal activity has been demonstrated in mouse cortical slices by Dunn and co-authors [17]. The mathematical model of neurovascular coupling, including all the above-mentioned mechanisms, was published by Kenny and co-authors [149]. They proved that the TRPV4-induced increase in [Ca^2+^]_i_ activates BK_Ca_ channels and thus contributes to the release of more K^+^ into the ECS, hence vasodilation. However, they stated that the release of Ca^2+^ and EET from astrocytes alone is not sufficient to activate BK_Ca_ channels, and the simultaneous depolarization of the astrocyte membrane is necessary to elicit a vascular response (Figure 2C).

Independently of neuronal activity, astrocytes can perceive direct mechanical stimulation and respond by increasing [Ca^2+^]_i_ when acutely reduced blood perfusion occurs [103,133,150]. Although the elucidation of astrocytic mechanisms leading to increased intracellular Ca^2+^ levels and subsequent production of vasoactive molecules is still under investigation, recent studies have indicated that the TRPV4 channel may be included. Due to being strategically localized in the endfeet that tightly enwrap the abluminal side of penetrating and intraparenchymal cerebral vessels, the TRPV4 channel is perfectly placed to sense the vascular tone. Kim and co-authors provided evidence that the parenchymal arteriole tone is regulated by a mechanism including astrocytic Ca^2+^ signals, triggered by TRPV4 activation. The increase in arteriolar flow/pressure led to a [Ca^2+^]_i_ rise in astrocytes and a subsequent vasoconstriction in cortical acute brain slices. Since the vasoconstriction was attenuated not only by the TRPV4 channel inhibitor but also by the antagonist of purinergic receptors, it is likely that purinergic signaling also plays a role in this process [8]. Similarly, Diaz and co-authors showed that angiotensin II-induced hypertension results in TRPV4 mediated astrocytic [Ca^2+^]_i_ changes and subsequent changes in perivascular arterial blood flow [114]. Another study by Haidey and co-authors contributed to the elucidation of the mechanism of TRPV4 channel opening, and the associated Ca^2+^ increase in astrocytes [150]. In mouse and rat cortical slices, they described a bi-directional communication between arterioles and the adjacent astrocyte endfeet mediated by TRPV4 channels and Ca^2+^-activated COX-1, which they propose plays a role in the mechanism of microvasculature oscillatory activity, namely very low-frequency cerebral blood flow oscillations. Sensitivity to direct mechanical stimulation and the ability to respond to changes in the vascular lumen diameter have also been defined in brainstem astrocytes, specifically in cardiovascular sympathetic control circuits, where they respond to a decrease in cerebral blood flow [133]. The authors described an interaction of astrocytic TRPV4 with connexin 43 (Cx43), resulting in ATP release from astrocytes and subsequent excitatory effects on the brainstem sympathetic control circuits. Thus, astrocytes not only affect local blood flow but can also influence systemic arterial blood pressure and heart rate to maintain blood flow through the brain (Figure 2D).

#### 4.3.4. Controlling Neuronal Synapse, Gliotransmitter Release

Astrocytes can modulate neuronal synapses via gliotransmitter releases, such as glutamate, ATP, or D-serine, in response to signals from neurons that induce changes in astrocytic [Ca^2+^]_i_ [151,152]; for review see Newman 2003 [153]. However, the exact mechanisms of this communication between astrocytes and neurons have not yet been fully elucidated. Shibasaki and co-authors proposed a mechanism of astrocyte activation through TRPV4 channels, resulting in the release of gliotransmitters and modulation of synaptic transmission [106]. This process involves the propagation of Ca^2+^ waves to neighboring astrocytes via gap junctions, and/or via ATP activation of astrocytic purinergic receptors.

#### 4.3.5. Cell Volume Regulation

Brain osmolality is under the strict control of astrocytes, which take up ions and neurotransmitters from the ECS and redistribute them along with water throughout the nervous tissue. Like most cells, astrocytes are continuously exposed to changes in the osmotic gradient due to metabolic activity, cell signaling, etc., to which they respond by changing their volume. Cells respond to these fluctuations by regulatory mechanisms called regulatory volume decrease and increase. Astrocytes, as the main homeostatic cells of the CNS, are often exposed to high concentrations of osmolytes, which they take up from the intercellular space and change their volume as a result [154]. The swelling of astrocytes is often associated with the development of tissue edema in various nervous system injuries, e.g., trauma, ischemia, brain tumors or hepatic encephalopathy. For this reason, the mechanisms of astrocyte swelling and the volume regulation have been extensively studied as possible targets for edema therapy. Changes in astrocyte volume activate sensors, which then trigger a cellular response that activates other downstream mechanisms. Cell volume regulation occurs through the activation of K^+^ and Cl^−^ ion channels, either through direct changes in cell volume or through Ca^2+^-dependent mechanisms. TRPV4 is one of the Ca^2+^-permeable channels sensitive to changes in cell membrane volume or stretch. Cell swelling is associated with increases in [Ca^2+^]_i_ resulting in the activation of several intracellular processes, including the stimulation of Ca^2+^-activated ion channels, the release of osmolytes, gliotransmitters, and AA.

TRPV4 was originally described as an osmotically activated channel and named accordingly: vanilloid receptor–related osmotically activated channel (VR-OAC) [2,155]. However, it was later found to respond to volume increase regardless of the molecular mechanism underlying the cell swelling, and therefore came to be considered as a sensor of volume rather than of osmotic changes [60]. Benfenati and co-authors proposed the cooperation of TRPV4 and AQP4 channels in hypoosmotically induced RVD and the formation of a functional TRPV4/AQP4 complex. Using the co-immunoprecipitation method, the authors noted the physical interaction of these two channels in rat cortical astrocytes. They also observed the swelling-induced Ca^2+^ influx through TRPV4 channels, which was abolished by the depletion of AQP4. Therefore, they speculated that these two channels cooperate in the process of RVD [118]. Similar results were obtained by Jo and co-authors in a mouse retina [113]. Nevertheless, these authors did not observe a physical interaction between AQP4 and TRPV4. Instead, they described the cooperation in which the water influx via AQP4 causes astrocyte swelling, which subsequently activates the mechanosensitive TRPV4 channel. The entry of Ca^2+^ through TRPV4 then triggers RVD via Ca^2+^-dependent K^+^ channels or volume-regulated anion channels (VRACs). Additionally, the membrane stretch can activate PLA2, whose activity leads to the production of AA and its subsequent conversion to EETs. EETs can then further activate TRPV4 channels. This mechanism was described in retinal Müller glia, which express both AQP4 and TRPV4 in the whole population [113]. In mouse primary astrocytes and the cell line DI TNC1, Mola and co-authors confirmed the contribution of TRPV4 to calcium influx induced by hypoosmotic stimuli but challenged its involvement in RVD [123]. Although they observed that gadolinium or ruthenium red reduced intracellular Ca^2+^ waves induced by the hypotonic stimulus, they found no changes in RVD kinetics. In more recent work, the same authors confirmed the molecular and functional interaction of AQP4 and TRPV4 in differentiated astrocytes. They found an upregulation of AQP4 in differentiated astrocytes, which was related to the increase in water permeability and more efficient regulatory volume processes. Accordingly, the magnitude of hypotonic-induced Ca^2+^ influx was doubled [156].

However, the results of experiments performed in acute brain slices are not consistent with in vitro results. In primary astrocyte cultures and in acute brain slices, Pivonkova and co-authors studied the role of astrocytic TRPV4 in swelling induced by hypoosmotic stress or oxygen-glucose deprivation (OGD) [68]. While they observed RVD in cultured astrocytes and confirmed that the TRPV4 channel is required for RVD, they did not detect this phenomenon in GFAP/EGFP astrocytes in acute brain slices. A likely explanation for these differences is the limited RVD in acute brain slices as well as in vivo, as also observed in other studies [157]. Another critical factor may be the severity of the stimulus [68,113]. Moreover, since TRPV4 expression appears to be restricted to certain subpopulations of astrocytes, brain regions, and specific cellular compartments such as perivascular processes, RVD may be restricted only to these sites. Toft-Bertelsen and co-authors tested the involvement of TRPV4 in volume regulation following the stimulus-evoked astrocytic swelling in acute hippocampal brain slices from rats. They observed changes in ECS volume, which serves as an indirect indicator of cell swelling, induced by the stimulation of synaptic activity, i.e., not under extreme non-physiological conditions such as hypoosmotic stress. The involvement of TRPV4 channels was tested using the non-selective TRPV4 inhibitor ruthenium red or the more-selective HC067047. The inhibition of TRPV4 channels slightly increased ECS shrinkage but did not affect RVD. The activation of TRPV4 with the agonist GSK1016790A had no significant effect on the ECS dynamics in hippocampal brain slices [107]. In contrast, Chmelová and co-authors described less ECS shrinkage in TRPV4 global knockouts compared to the control during exposure to OGD [158], which represent more severe conditions. This result suggests the involvement of the TRPV4 channel in the swelling of cerebral neuropil cells, but the contribution of each cell type cannot be distinguished. Consistent with these data, Hoshi and co-authors observed reduced OGD-induced swelling in TRPV4 knockouts or after treatment with the inhibitor HC-067047 and, conversely, increased swelling after TRPV4 channel activation with the agonist GSK1016790A [80].

#### 4.3.6. The Role of Astrocytic TRPV4 in the Brain Ischemia

Astrocytes play a dual role in ischemic brain injury. In the acute phase of ischemia, they release inflammatory molecules that may worsen ischemic damage. Conversely, they may behave as anti-excitotoxic, protecting neurons from oxidative stress [159,160] and producing neuroprotective neurotrophins [161]. Moreover, astrocytes contribute significantly to the development of cerebral edema, which can be not only a serious complication of the ischemic stroke but often a cause of death. This results from an increased uptake of osmotically active compounds, including glutamate, K^+^ ions or lactate [162]. Astrocyte functional integrity is also crucial for the formation of the BBB, whose disruption typically occurs in the acute phase of ischemia [163]. Finally, astrocytes protect neurons from excitotoxicity caused by excessive glutamate release during ischemia [164,165]. In the later recovery phase, astrocytes play a dual role again. Although astrocytes stimulate neurogenesis, synaptogenesis and angiogenesis, and promote neurological recovery, they also form a glial scar, which obstructs the growth of axons and thus the restoration of neuronal function in the affected area [166]. However, the glial scar forms a physical barrier between healthy tissue and an ischemic lesion that prevents further spread of tissue damage and protects cells against toxic molecules spreading from the infarct core [167,168].

Astrocytic TRPV4 has been shown to be involved in a couple of Ca^2+^ activated processes of the ischemic cascade–excitotoxicity [15,169], peri-infarct depolarization [170], oxidative stress [6] or astrocytic swelling [113,118].

Peri-infarct depolarization (PID) is caused by ionic imbalance and is generated in the ischemic core, spreading within the peri-infarct area (penumbra). It has a major impact on the progression of ischemic damage towards healthy tissue and the resulting extent of damage (infarct volume). It is accompanied by a cellular Ca^2+^ overload in astrocytes and neurons, which triggers a cascade of cellular events resulting in apoptosis, resulting in the release of glutamate and other excitatory amino acids in the ECS. Ca^2+^ signaling during PID in astrocytes has been shown to be primarily mediated by the release from intracellular stores, but other pathways, such as TRPV4 channel activation, are also likely to be involved [170]. Rakers and co-authors showed that during PID changes in intracellular Ca^2+^, concentrations occur in both astrocytes and neurons, with the subsequent release of glutamate. Concentration changes were influenced by the TRPV4 function, which was confirmed using both knockouts and the specific inhibitor HC-067047. However, the TRPV4 deletion or inhibition did not affect the threshold and overall burden of PIDs [170].

Oxidative stress, resulting from the disbalance of production of free radicals and the capacity of the antioxidant systems, is a key factor in the initiation of ischemia/reperfusion brain injury [171]. The role of astrocytic TRPV4 in oxidative stress has been proposed by Bai and co-authors [6], who observed increased astrocyte viability in organotypic hippocampal culture during oxidative stress, evoked by mercaptosuccinate or buthionine sulfoximine, after the inhibition of the TRPV4 channels by Gd^3+^ or Ruthenium red.

Most attention has so far been paid to the TRPV4 channel in terms of its involvement in cell volume regulation and cerebral edema formation. Its potentially protective role in cerebral edema is related to its involvement in RVD, as suggested in vitro [113,118]. However, the results of in vivo experiments are so far rather inconclusive. Jie and co-authors showed that pharmacological inhibition of TRPV4 by intracerebroventricularly administered HC-067047 is neuroprotective and resulting in less brain edema after transient middle cerebral artery occlusion (tMCAO) [21]. They proposed that the block of TRPV4 may inhibit the activation of metalloproteinases, which helps to reduce the disruption of BBB integrity and the subsequent vasogenic brain edema. Pivonkova and co-authors used global TRPV4 knockout to examine the role of this channel in ischemia-induced cerebral edema [68]. Ischemia was modeled using permanent middle cerebral artery occlusion (pMCAO) and the edema was quantified using T2-weighted magnetic resonance imaging (MRI). Twice as large edema was observed one day after pMCAO in TRPV4^−/−^ mice when compared to the controls, clearly confirming the role of the TRPV4 channel in the development of cerebral edema. However, the authors did not confirm the contribution of TRPV4 channels to the astrocytic RVD and observed a very low TRPV4 channel expression in the astrocytes sorted based on GFAP or GLAST expression. Therefore, the authors believe that the effect of TRPV4 deletion on the extent of cerebral edema was not solely at the level of astrocyte function. On the contrary, 24 h after tMCAO, Tanaka and co-authors observed a smaller ischemic lesion as well as smaller water content in TRPV4^−/−^, compared to the controls [22]. This observation was associated with vasoconstriction caused by swelling of the astrocytic processes. The authors explain the discrepancy between their results and those of Pivonkova and co-authors by using different models of focal cerebral ischemia, namely transient and permanent, respectively. They suggest that pMCAO may exacerbate the damage observed in TRPV4 knockouts, due to the lack of development of collateral circulation. Another explanation may be the way postischemic damage was quantified. While the team of Tanaka et al. assessed the lesion size based on staining with 2,3,5-Triphenyltetrazolium Chloride (TTC), Pivonkova and co-authors quantified the overall lesion size from T2-weighted MRI images. However, greater damage in TRPV4^−/−^ was also confirmed by immunohistochemical labeling of GFAP and Iba1 [69]. In addition, different strains of mice were used in the two studies. Finally, in the same TRPV4^−/−^ mouse strain used in the publication by Pivonkova and co-authors, no difference was observed between the control and knockout at day 3 after tMCAO, in either neurological score, lesion size (TTC staining), or regional blood flow [86].

In a mouse model of intracerebral hemorrhage (ICH) induced by intrastriatal injection of collagenase, activation of the TRPV4 channel with a selective agonist GSK1016790A resulted in improved neurological and motor deficits [172].

## 5. Conclusions

In this review, we summarize the current knowledge of the multifunctional and broadly expressed TRPV4 channel. This channel is present in all cells of the nervous system, i.e., neurons, glia, endothelial, and smooth muscle cells. In addition, since it is activated by various physical and chemical stimuli, it plays a role in various physiological and pathological cellular processes, including proliferation, glial activation, osmoregulation, and oxidative stress. As such, it is a key player in a range of neurological diseases, including aging, neurodegenerative diseases, stroke, epileptic seizures, pain, tumors, and trauma. In the pathology of ischemic brain injury, the different and competing roles played by TRPV4 channel activation in the brain can be amply demonstrated. Since TRPV4 channels are permeable to Ca^2+^, their activation can induce different cellular events, and the resulting Ca^2+^ influx can have opposite consequences, most likely depending on the cell type and the expression of other effectors in the cell itself and in neighboring cells. With strokes, it can result in excitotoxicity, peri-infarct depolarization, glial cell activation, and neuroinflammation on the one hand and volume regulation and restoration of blood flow on the other. The activation of TRPV4 channels in neurons during cerebral ischemia enhances glutamate release and excitotoxicity, but it subsequently helps to replace the damaged neurons with new ones. Although astrocytic TRPV4 channels play a role in oxidative stress and contribute to PID, they may also regulate cell volume and prevent the development of cerebral edema. In addition, together with the TRPV4 of endothelial and smooth muscle cells, astrocytic TRPV4 channels help to restore cerebral blood flow during oxygen/glucose insufficiency. However, endothelial TRPV4 may also promote neuroinflammation and initiate BBB disruption. Finally, TRPV4 in microglia plays a role in the modulation of the migratory ability of these cells and in the process of activation. However, it is uncertain whether its activation promotes the pro- or anti-inflammatory phenotype of microglia.

For all these reasons, the results of using TRPV4 channel modulators in mouse models of cerebral ischemia are highly inconsistent. It has also been shown that channel expression varies significantly in different brain regions and that only certain subpopulations of glial cells express a functional TRPV4 channel. Finally, it is also important to note that this channel is present in tissues of various body organs and therefore the potential side effects of TRPV4 channel modulation for the treatment of cerebral ischemia should be carefully evaluated. It is therefore essential to further address this issue intensively and to explain the spatial differences in expression, function, and regulation of these channels to achieve a more comprehensive understanding of the role of TRPV4 in stroke, which will ultimately lead to the development of new therapeutic strategies. 

## Figures and Tables

**Figure 1 ijms-24-07101-f001:**
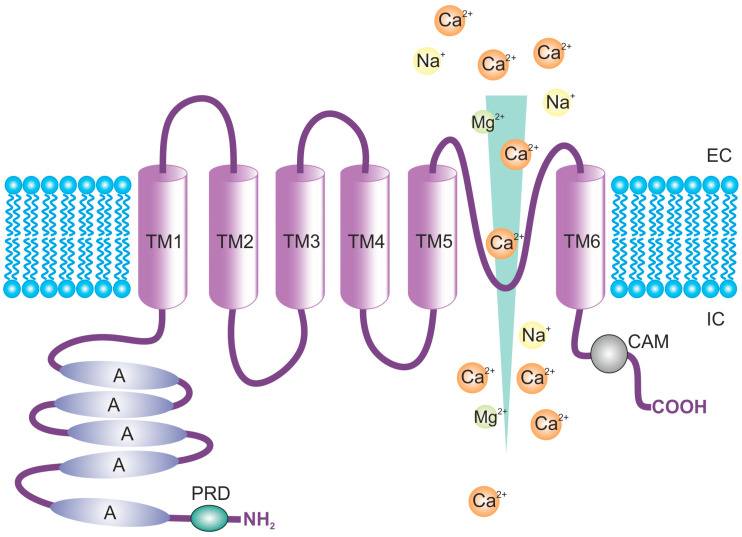
Structure of the transient receptor potential cation channel subfamily V member 4 (TRPV4), with the main functional domains. Abbreviations: A, ankyrin; CAM, Ca^2+^/calmodulin binding site; EC, extracellular side; IC, intracellular side; TM, transmembrane domain; PRD, proline-rich domain.

## Data Availability

Not applicable.

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
