# Peer review of "Astrocytic TRPV4 Channels and Their Role in Brain Ischemia"

_ijms, 2023, doi:10.3390/ijms24087101_

Round 1
Reviewer 1 Report
In the current manuscript, Tureckova et. al review the existing knowledge of TRPV4 channels with a specific focus on brain damage due to ischemia. Overall, it is a well organized and well-written semi-comprehensive review, although there is insufficient discussion about the role of TRPV4 channels in the brain. The authors should at least mention the inherited diseases of the brain, related to mutations in this channel e.g. skeletal dysplasias and neuropathies as the underlying biology of the LoF variants can be useful for understanding the role of TRPV4 in regulating brain function. The topic of pleiotropy and the spectrum of phenotypes associated with TRPV4 channels on account of multiple substrates as well as multiple stimuli should also be addressed a bit more, as it is relevant to the topic of disease and/or ischemia development.
Author Response
Responses to Reviewer 1
In the current manuscript, Tureckova et. al review the existing knowledge of TRPV4 channels with a specific focus on brain damage due to ischemia.
Point 1:
Overall, it is a well organized and well-written semi-comprehensive review, although there is insufficient discussion about the role of TRPV4 channels in the brain.
Response 1:
Based on the reviewer's comment, we have added an assessment of the general role of the TRPV4 channel in the brain to the Introduction and Conclusions sections.
Point 2:
The authors should at least mention the inherited diseases of the brain, related to mutations in this channel e.g. skeletal dysplasias and neuropathies as the underlying biology of the LoF variants can be useful for understanding the role of TRPV4 in regulating brain function.
Response 2:
We agree with the reviewer's comment that the role of TRPV4 in neurological disease is an exciting topic that is not fully covered in our paper. However, because this topic is very broad, we chose to focus narrowly in our publication on the role of astrocytes in ischemic brain injury, a topic that has not been fully explored and, so far, a number of conflicting results have been published.
However, a paragraph focusing on naturally occurring mutations and genetic diseases associated with the TRPV4 channel has been added to the chapter on the structure of the TRPV4 channel.
Point 3:
The topic of pleiotropy and the spectrum of phenotypes associated with TRPV4 channels on account of multiple substrates as well as multiple stimuli should also be addressed a bit more, as it is relevant to the topic of disease and/or ischemia development.
Response 3:
Although we believe that most of the functions that TRPV4 channels have in the brain are mentioned in the text, we agree with this suggestion and have included paragraphs in the Introduction, Conclusions, and Chapters 3 and 4, that comment more on the broad expression and multimodality of TRPV4 channels with respect to their function in stroke pathology and other neurological diseases; highlighted in yellow.
Reviewer 2 Report
The authors of this manuscript discuss the role of TRPV4 non-selective cation channels and their involvement in the modulation of neuronal excitability, blood flow control and cerebral edema formation. The authors report the important role of the TRPV4 channel in mediating and regulating the flow of Ca2+ ions in neural cells in response to various stimuli, even during the process of cerebral ischemia, making this role possible therapeutic targets.
The authors report an analysis of the data available in the literature and describe all the aspects related to the TRPV4 channel, agonists, antagonists and its expression both in pathological and healthy conditions.
the authors summarized to highlight the scientific evidence to understand the interactomes of the TRPV4 channel with other channels to mediate several functions including volume regulation, neurotransmission, and other aspects. For this aspect I suggest to the authors to broaden the interactomes of the TRPV4 channel and to hit the target to highlight the role of TRPV4 as a therapeutic target.
Furthermore, I suggest adding ATP-sensing K+ (KATP) channels (including KCNJ8, KCNJ11, ABBC8 and ABCC9 genes) among the TRPV4 channel interactomes as TRP channels together with KATP and BK channels have a synergistic role in the regulation and the modulation of neuronal excitability, in the control of blood flow and in the formation of cerebral edema, I recommend expanding the bibliography with works dealing with the topic, for example:
1- F. Maqoud, R. Scala, M. Hoxha, B. Zappacosta, and D. Tricarico, “ATP-sensitive potassium channel subunits in neuroinflammation: new drug targets in neurodegenerative disorders.” CNS Neurol. I disturb. Drug Targets, January 2021, doi: 10.2174/1871527320666210119095626
I would recommend a thorough grammar check of the entire manuscript and a weighted edition of the introduction if too much information is presented in a non-fluid and on-point manner.
Author Response
Responses to Reviewer 2
The authors of this manuscript discuss the role of TRPV4 non-selective cation channels and their involvement in the modulation of neuronal excitability, blood flow control and cerebral edema formation. The authors report the important role of the TRPV4 channel in mediating and regulating the flow of Ca2+ ions in neural cells in response to various stimuli, even during the process of cerebral ischemia, making this role possible therapeutic targets.
The authors report an analysis of the data available in the literature and describe all the aspects related to the TRPV4 channel, agonists, antagonists, and its expression both in pathological and healthy conditions.
Point1:
the authors summarized to highlight the scientific evidence to understand the interactomes of the TRPV4 channel with other channels to mediate several functions including volume regulation, neurotransmission, and other aspects. For this aspect I suggest to the authors to broaden the interactomes of the TRPV4 channel and to hit the target to highlight the role of TRPV4 as a therapeutic target.
Response 1:
The chapter on the interaction of TRPV4 channels with other astrocytic proteins has been expanded. Potassium channels Kir, K-ATP, and BKCa as well as glutamate receptors have been included. The cooperation of these proteins is discussed in relation to their role in cell volume regulation, regulation of cerebral blood flow, and excitotoxicity.
Point 2:
Furthermore, I suggest adding ATP-sensing K+ (KATP) channels (including KCNJ8, KCNJ11, ABBC8 and ABCC9 genes) among the TRPV4 channel interactomes as TRP channels together with KATP and BK channels have synergistic role in the regulation and the modulation of neuronal excitability, in the control of blood flow and in the formation of cerebral edema, I recommend expanding the bibliography with works dealing with the topic, for example:
Maqoud, R. Scala, M. Hoxha, B. Zappacosta, and D. Tricarico, “ATP-sensitive potassium channel subunits in neuroinflammation: new drug targets in neurodegenerative disorders.” CNS Neurol. I disturb. Drug Targets, January 2021, doi: 10.2174/1871527320666210119095626
Response 2:
Relevant citations on the expression and function of ATP-sensing potassium channels in astrocytes were included.
Point 3:
I would recommend a thorough grammar check of the entire manuscript and a weighted edition of the introduction if too much information is presented in a non-fluid and on-point manner.
Response 3:
The introductory section has been edited to improve the clarity, flow and readability of the text.
The manuscript was revised and grammar checked by a native speaker.
Round 2
Reviewer 2 Report
The authors of the manuscript have modified and improved the introduction section and made it more clear and the objectives are very clear.
Furthermore , the authors have enlarged the whole part of the TRPV4 channel interatome and have highlighted the synergistic role in the regualtion of all the aspects mentioned in the manuscript.